# Multiple resolution residual network for automatic thoracic organs-at-risk segmentation from CT

**Hyemin Um**[1] **Jue Jiang**[1] **Maria Thor**[1] **Andreas Rimner** [1] **Leo Luo** [1] **Joseph O. Deasy**[1] **Harini Veeraraghavan**[1]

VEERARAH@MSKCC.ORG [1] *Department of Medical Physics, Memorial Sloan-Kettering Cancer Center, New York, NY 10065, USA*

## Abstract

We implemented and evaluated a multiple resolution residual network (MRRN) for multiple normal organs-at-risk (OAR) segmentation from computed tomography (CT) images for thoracic radiotherapy treatment (RT) planning. Our approach simultaneously combines feature streams computed at multiple image resolutions and feature levels through residual connections. The feature streams at each level are updated as the images are passed through various feature levels. We trained our approach using 206 thoracic CT scans of lung cancer patients with 35 scans held out for validation to segment the left and right lungs, heart, esophagus, and spinal cord. This approach was tested on 60 CT scans from the open-source AAPM Thoracic Auto-Segmentation Challenge dataset. Performance was measured using the Dice Similarity Coefficient (DSC). Our approach outperformed the best-performing method in the grand challenge for hard-to-segment structures like the esophagus and achieved comparable results for all other structures. Median DSC using our method was 0.97 (interquartile range [IQR]: 0.97-0.98) for the left and right lungs, 0.93 (IQR: 0.93-0.95) for the heart, 0.78 (IQR: 0.76-0.80) for the esophagus, and 0.88 (IQR: 0.86-0.89) for the spinal cord.

**Keywords:** Multiple residual feature streams, thoracic normal organs, AAPM thoracic grand challenge dataset.

## 1. Introduction

Precise tumor targeting while reducing unnecessary dose to critical normal organs, especially those within the mediastinum like the esophagus, using high-dose radiation therapy treatments require highly accurate segmentations (Mackie et al., 2003). Clinically used manual delineations are both time consuming and highly variable (Yang et al., 2012). Therefore, we implemented a deep learning-based automatic OAR segmentation method.

Typical methods for thoracic CT OAR segmentations include 2D/3D U-Net architectures (Feng et al., 2019) as used in the the 2017 AAPM thoracic CT grand challenge (Yang et al.). Recently, (Dong et al., 2019) combined generative adversarial networks with fully convolutional network (FCN) discriminators. Although FCNs have been successfully applied for several medical image segmentation and computer vision tasks (Zhou et al., 2017; Pan et al., 2017; Long et al., 2015), segmenting small structures remains a challenging task for current methods due to the loss of resolution in the deeper layers.

Approaches to solve the issue of loss of resolution employ dilated convolutions (Chen et al.,

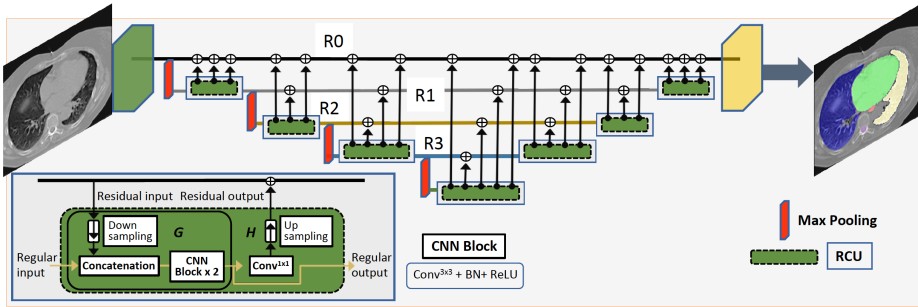

Figure 1: Multiple resolution residual network. Multiple residual feature streams R0, R1, R2, R3 are shown. Convolutional blocks are composed of a sequence of convolutions, batch normalization and ReLU activations.

2018), pyramidal pooling (Zhao et al., 2017), multi-stage hierarchical networks (Roth et al., 2017), and organ attention modules (Wang et al., 2019). Our approach builds on these methods by combining information from multiple feature streams computed at different image resolution levels that preserve different levels of spatial context for segmentation.

## 2. Method

The architecture of our network is summarized in Figure 1. As shown, our network computes multiple feature streams after each image pooling operation. These feature streams (eg. R0, R1, R2, R3 in Fig. 1) carry features computed at a particular image resolution and provide contextual information at a higher image resolution when residually combined with deeper layer features. The feature streams are both residually input to every feature layer and modified after passing through each layer through reverse connections (black arrows in Fig. 1). Therefore, features in the deeper layers combine the inputs from the immediately preceding layer and the modified feature streams to compute new features. Such residual feature combination was done to improve stability of training a deep network (He et al., 2016). Also, features from all available feature resolutions are used in each feature layer. This connection strategy is different from and improves over skip connections used in the more common U-Net architectures (Ronneberger et al., 2015), where only features at a particular image resolution from the encoder are directly concatenated with the corresponding decoder layer features. It is also different from residual connections used in ResNet architectures (He et al., 2015), where only the immediately preceding feature layer input is connected to a given feature layer.

Features processing is done in the residual connection units (RCU). RCU takes in a residual input from one of the preceding higher resolution feature streams (after appropriate down-sampling) and features computed from the immediately preceding CNN layer or a RCU within a RCU block. These two features are channel-wise concatenated and processed through one or more CNN blocks (Fig. 1). The output of a RCU consists of a residual output that is passed back to the feature stream after 1×1 convolution of the regular output. Regular output or the feature map is passed to the next RCU or the next CNN layer. A RCU block consists of one or more RCU units. The convolutional or CNN block used in a RCU (Fig. 1) is composed of 3×3 convolutions, batch normalization (BN) and ReLU activation. This approach helps to alleviate the issue of down-sampling and enlarges semantic

Table 1: DSC achieved for thoracic OARs in the AAPM online testing set.

| Method | 2D/3D | Left Lung | Right Lung | Heart | Esophagus | Spinal Cord |
|--------|-------|-----------|------------|-------|-----------|-------------|
| MRRN | 2D | 0.96 ± 0.01 | 0.96 ± 0.02 | 0.93 ± 0.03 | 0.77 ± 0.04 | 0.87 ± 0.017 |
| Elekta | 2.5/3D | 0.97 ± 0.02 | 0.97 ± 0.02 | 0.93 ± 0.02 | 0.72 ± 0.10 | 0.88 ± 0.037 |
| UVa | 3D | 0.98 ± 0.01 | 0.97 ± 0.02 | 0.92 ± 0.02 | 0.64 ± 0.20 | 0.89 ± 0.042 |
| Mirada | 2D | 0.98 ± 0.02 | 0.97 ± 0.02 | 0.91 ± 0.02 | 0.71 ± 0.12 | 0.87 ± 0.110 |

context, which is important for segmenting small or long and narrow structures like the spinal cord or esophagus.

**Implementation details:** The MRRN network was implemented using the Keras and Tensorflow library (Chollet et al., 2015) and trained on Nvidia GTX 1080Ti with 12GB memory. The network was optimized using the ADAM algorithm (Kingma and Ba, 2014) with an initial learning rate of 1e-4 and cross-entropy loss for 50 epochs with a batch size of 10. The network consisted of 28,941,717 trainable parameters. The model producing the best overall average DSC across all five structures on the validation set was selected.

## 3. Datasets

Two independent datasets were used for the analysis. The thoracic CT scans of 241 patients with locally advanced non-small cell lung cancer (LA-NSCLC) imaged at our institution comprised the internal cohort used for training (N=206) and validation (N=35). The 2017 AAPM Thoracic Auto-Segmentation Challenge (Yang et al.) contains 60 CT scans from 3 different institutions. Of these, 36 were provided for training, 12 for offline testing, and 12 for online testing. We used all 60 cases for testing, wherein the training and offline cases (N=48) formed testing set 1 and the online (N=12) cases belonged to testing set 2.

## 4. Experiments and evaluation metrics

Training and validation were performed in 2D with a total of 21441 and 2104 images, respectively, of size 256×256. Testing set 1 was used to evaluate the accuracy of the generated segmentations when compared against expert delineations using the DSC. Additionally, the performance of our method was benchmarked against the 3 best-performing methods in the AAPM grand challenge using testing set 2. These methods include the U-Net and the VGGNet (Simonyan and Zisserman, 2014).

## 5. Results

Our method resulted in the median DSC of 0.97 (IQR: 0.97-0.98) for the left and right lungs, 0.93 (IQR: 0.93-0.95) for the heart, 0.78 (IQR: 0.76-0.80) for the esophagus, and 0.88 (IQR: 0.86-0.89) for the spinal cord in testing set 1. Table 1 shows a comparison of the segmentation accuracy achieved by the various methods expressed as mean DSC ± standard deviation for the five structures in testing set 2. Our method, MRRN, produced the most accurate segmentations for the esophagus. The accuracy was comparable between the four methods for the remaining analyzed organs. Figure 2 shows a representative case from the online testing phase of the 2017 AAPM grand challenge. As shown by the yellow-colored overlap between the two masks, our method produced a reasonably accurate segmentation for all five structures.

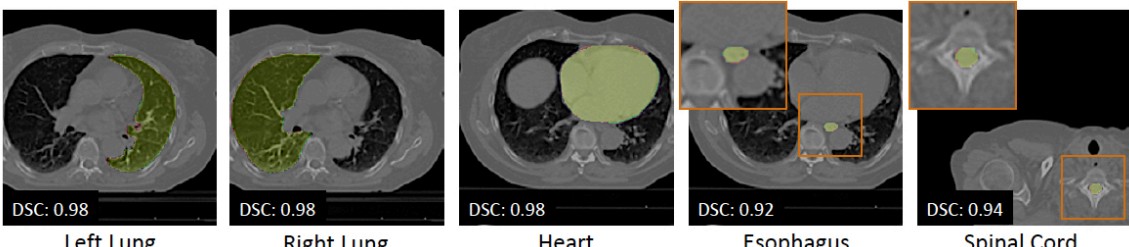

Figure 2: Example segmentations for the analyzed organs. The red masks correspond to algorithm segmentations and green masks to expert delineations.

## 6. Conclusion

We evaluated a multiple resolution residual network for segmenting normal organ structures from thoracic CT scans. Our method showed performance improvements over existing methods.

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
