# OpenReview forum: "Multiple resolution residual network for automatic thoracic organs-at-risk segmentation from CT"
_MIDL.io/2020/Conference — MIDL 2020_

### Official Review · AnonReviewer3 · 2020-03-10
**this paper aims to describe Multiple resolution residual network for automatic thoracic organs-at-risk segmentation from CT**

**Rating:** 3
**Confidence:** 3

**Review:**

The paper is very well organized. This is a well written paper and everything is clear.
Paper is easy to follow with clear motivation about the method.
The dataset used in this study is large, from multiple sites and the performance of their segmentation network is interesting. However the weaknesses of this paper is that the authors didn't mention the effect of different reconstruction kernels, slice thickness or the effect of contrast injection in their model.

---

### Official Review · AnonReviewer2 · 2020-03-12
**multi-scale method for organ segmentation**

**Rating:** 2
**Confidence:** 3

**Review:**

#Summary
This work proposed a new deep-learning architecture for the segmentation of normal organs at risk in thoracic CT data. The authors introduce residual connections from downed scale to upper scales for skip connection of U-Net architecture. They explained these residual connections between down-scaled feature map and upper feature maps achieve multi-resolution feature learning of volumetric data.

#Pros
Performance comparison among the proposed method and the three best-performance methods in the AAPM grand challenge by using two independent datasets for train and test, respectively.
-	The proposed method achieved 0.05-0.13 higher segmentation accuracy (dice similarity coefficient) than the other three methods for esophagus segmentation.
-	For the other organs except esophagus, the proposed method achieved 0.01-0.02 less or equal segmentation performances than other methods. It looks comparable.

#Cons
No theoretical and reasonable explanations about how to select the connection path in Fig. 1 There are several options to connect different scales.

In Fig.1, the paths from down-scaled features to upper-scale features exist even in up-convolution parts of U-Net. It looks strange for me, because feature extraction might be done in encoding part, that is, the former part of U-Net before up convolutions. Why?

What kind of operations the architecture adopted is unclear for the handling of the different size of feature in residual connection. How to upsample is not presented. Just zero padding, nearest neighbor interpolation, bilinear or cubic interpolation, or Gaussian pyramid?

---

### Official Review · AnonReviewer1 · 2020-03-13
**Well-written, lacking a bit of technical detail**

**Rating:** 3
**Confidence:** 4

**Review:**

This paper presents a unet-like architecture that is enriched with skip connections from lower levels of the downsampling/upsampling paths towards upper levels. The task the authors attempt to solve is multi-organ segmentation from CT scans, and results are comparable or better than the state-of-the-art, according to a nice evaluation (the test set of a grand-challenge). I believe this is a solid short paper and I support acceptance. I would like however to see more technical details about what is the exact way in which connections are built in this network. Figure 1 could benefit from a better written caption, in this sense.

Minor comments:
1) In the first page, you probably wanted to write "generative" instead of "generational"?
2) Could you please clarify if the input/output of your architecture is volumetric or bidimensional? You first mention that you implemented a volumetric OAR segmentation method, but later in the text you say you had 21,000 256x256 images, which sounds like you dealt with 2d images.

---

### Official Review · AnonReviewer4 · 2020-03-14
**Decent approach for unconvincing results**

**Rating:** 2
**Confidence:** 5

**Review:**

1. Overall, the approach lacks originality, as multi-resolution feature fusion has been explored by the deep learning community.

2. It is difficult to see 2 inputs and 2 outputs in figure 1 as mentioned in section 2.

3. The fourth pooling layer is not connected to any layer/block in figure 1.

4. It would be better to have names of the authors along with the method names in table 1.

5. The proposed approach is best only for the segmentation of esophagus and not for other organs.

6. It is not clear whether the word "images" in section 4 denotes 2D slices or 3D scans.

---

### Meta-Review · Area_Chair1 · 2020-04-06
**MetaReview of Paper308 by AreaChair1**

**Rating:** 3

**Metareview:**

Strengths: This paper is very well organized and written. The proposed approach is not novel, but is quite decent - the contribution is a well-validated application. A public dataset is used with comparison w.r.t. other approaches.

Weaknesses:
- There is a lack of technical details on how the residual connections are build, on the input/output size, on the upsampling path.
- Out of 5 organs, the proposed approach is superior for only one organ. For the rest, it seems comparable to the other methods. So the results are reasonable, however not outstanding.

The « Weak Reject » ratings are based mainly on the lack of technical details and lack of originality. Regarding lack of originality, the paper is a well-validated application and acknowledged as such by the authors. Regarding the lack of technical details, it is also noted by the reviewers who accepted the paper.

Since the major argument in favor of weak reject is the lack of details - which I believe can be addressed by the authors - my recommendation is towards a weak acceptance of this paper.

**Paper Type:**

validation/application paper

---

### Decision · Program_Chairs · 2020-04-11

Accept